# Roles of Essential Oils, Polyphenols, and Saponins of Medicinal Plants as Natural Additives and Anthelmintics in Ruminant Diets: A Systematic Review

**DOI:** 10.3390/ani13040767

**Published:** 2023-02-20

**Authors:** Diky Ramdani, Endah Yuniarti, Anuraga Jayanegara, Abdul Shakoor Chaudhry

**Affiliations:** 1Department of Animal Production, Faculty of Animal Husbandry, Jatinangor Campus, Universitas Padjadjaran, Sumedang 45363, Indonesia; 2Department of Animal Nutrition and Feed Technology, Faculty of Animal Husbandry, Jatinangor Campus, Universitas Padjadjaran, Sumedang 45363, Indonesia; 3Department of Animal Nutrition and Feed Technology, Faculty of Animal Science, IPB University, Bogor 16680, Indonesia; 4School of Natural and Environmental Sciences, Newcastle University, Newcastle upon Tyne NE1 7RU, UK

**Keywords:** essential oils, polyphenols, saponins, feed additives, anthelmintics, ruminants

## Abstract

**Simple Summary:**

Ruminant nutritionists have been challenged to improve animal production efficiently but at the same time produce healthy and environment friendly ruminant-derived food products. Recent studies on utilizing essential oils, polyphenols, and saponins of herbal plants show that these bioactive components can play important roles as alternative natural dietary additives and anthelmintics, in order to replace growth-promoting antibiotic and chemical anthelmintic treatments. Since the prohibition of using growth-promoting antibiotics and chemical anthelmintics, the global market has emphasized the use of natural feed additives and anthelmintics as alternatives for ruminants. This article presents the potentials and problems of using plant-based bioactive compounds for sustainable ruminant diets to support food safety and food security.

**Abstract:**

Public awareness on health and safety issues in using antibiotics for livestock production has led many countries to ban the use of all growth-promoting antibiotics (GPA) for livestock feeding. The ban on the utilization of antibiotics in livestock, on the other hand, is an opportunity for researchers and livestock practitioners to develop alternative feed additives that are safe for both livestock and the consumers of animal derived foods. Many feed additives were developed from a number of plants that contain secondary metabolites, such as essential oils, polyphenols, and saponins. These secondary metabolites are extracted from various parts of many types of plants for their uses as feed additives and anthelmintics. Recent investigations on using essential oils, polyphenols, and saponins as dietary additives and anthelmintics demonstrate that they can increase not only the production and health of ruminants but also ensure the safety of the resulting foods. There are many publications on the advantageous impacts of dietary plant bioactive components on ruminants; however, a comprehensive review on individual bioactive constituents of each plant secondary metabolites along with their beneficial effects as feed additives and anthelmintics on ruminants is highly required. This current study reviewed the individual bioactive components of different plant secondary metabolites and their functions as additives and anthelmintics to improve ruminant production and health, with respect to safety, affordability and efficiency, using a systematic review procedure.

## 1. Introduction

Public awareness surrounding the health and safety issues of using antibiotics for livestock production, including ruminants, has led many countries such as the EU to ban all growth-promoting antibiotics (GPA) in livestock feeding [1,2]. The prohibition of using GPAs in food animal diets has also been applied in Indonesia by the Regulation No. 14/2017 of Indonesian Ministry of Agriculture [3]. Increased level of GPA use to improve ruminant production may lead to more residues of antibiotics in meat, milk and manure that cause the occurrence and possible transmission of antibiotic-resistant bacteria to humans and the environment [2,4].

Exploiting indigenous plants rich in secondary metabolites for their use as safe additives to replace GPAs in ruminant diets is preferable, since the public expects to consume more healthy and sustainable meat and milk products [4]. Many plants produce secondary metabolites as bioactive constituents to protect them against bacterial, fungal, or insect predators, but they are not primarily involved in the main biochemical processes such as growth and reproduction [4,5,6]. Essential oils, phenols, tannins and saponins are highly prospective natural dietary additives for their use to modify rumen functions, enhance protein and energy use [6,7], reduce methane (CH4) production [8,9] and improve meat and milk qualities [10,11]. Plant bioactive compounds can also be utilized as health-promoting additives in ruminant diets to control bloat and nematodes [3,12,13].

The utilization of medicinal plants as natural feed additives and anthelmintics to optimize ruminant production would be highly dependent upon the types of bioactive constituents. The optimum dose and feeding duration should be taken into account when using plant bioactive-based additives in ruminant diets [3,14]. A comprehensive review study on individual bioactive constituents of each set of plant secondary metabolites and their multiple efficacies as feed additives and anthelmintics in ruminant diets is not yet widely available. Therefore, this study systematically reviews and discusses the potentials and problems of using specific bioactive constituents of selected naturally available plants as dietary additives and anthelmintics for ruminants.

## 2. Methods

### 2.1. Literature Research

The literature database consisted of published articles in internationally reputable scientific journals from 2000 to 2022. The articles were searched for on several scientific platforms, such as Science Direct, Scopus, and Google Scholar. The search on each platform used Boolean Operators, where the keywords consisted of: “plant” AND (“Essential oils” OR “Polyphenols” OR “Saponins”) AND (“Cattle” OR “sheep” OR “goat”) AND “Performance “ AND “Anthelmintics”. The results were stored and integrated with a reference manager application (Mendeley desktop software version 1.19.8, Mendeley Ltd., Elsevier B.V.) for data selection purposes.

### 2.2. Inclusion Criteria and Selection Process

About 804 articles were initially selected and screened to ensure that their quality and relevance met the inclusion criteria for a systematic review. The first step was to check the database of articles in the reference manager software for their duplication potential in different databases. About 670 (83.3%) documents of the collected articles were not duplicated. The next steps of selection were based on the relevance of each title and abstract (selection 1), as well as on the full text and the reputation of each publisher (selection 2). Around 582 articles were excluded and about 88 articles from years of 2000 to 2022 met the inclusion criteria and further used for their appraisal in this systematic review as can be seen from Table 1, Table 2, Table 3, Table 4, Table 5, Table 6, Table 7, Table 8 and Table 9. Only two articles that were published earlier than 2000 [15,16] were used as references in this paper, because of their highly relevant contents. All selection processes for the eligibility of the above selected articles were done following the Preferred Reporting Items for Systematic Review and Meta-Analysis (PRISMA) procedure [17], as explained in Figure 1.

## 3. Results

### 3.1. Essential Oils Sources and Types

Essential oils (EO), recognzed as volatile oils, are commonly derived from edible, medicinal, herbal, or spice plants. The main plant tissues for EO deposition vary across the plants. They can be the leaves, flowers, stem, seeds, roots, rhizomes, or barks. The EO deposits are mostly extracted by using either steam distillation, hydro distillation, or organic solvent extractions [18]. The EO compounds are chemically a mixture of terpenoids, majorly monoterpenes (C10, about 90% EO content) and sesquiterpenes (C15), but they may contain diterpenes (C20) and numerous low-molecular-weight aliphatic hydrocarbons, acids, alcohols, aldehydes, acyclic esters, or lactones, as well as non-nitrogenous and sulphur-containing compounds [6,18,19].

**Table 1 animals-13-00767-t001:** Major compounds of selected essential oils in different botanical fractions of various plants.

Essential Oils	Scientific Names	Main Parts	Major Compounds	References
Anise oil	*Pimpinella anisum* L.	Fruits	(%) *trans*-anethole (76.9–93.7), *γ*-himachalene (0.4–8.2), *trans*-pseudoisoeugenyl 2-methylbutyrate (0.4–6.4), *p*-anisaldehyde (*trace*-5.4), methylchavicol (0.5–2.3).	[20]
Basil oil	*Ocimum basilicum* L.	Leaves, flower	(%) Estragole (52.6–58.3), limonene (13.6–19.4), fenchone (5.7–10.1), exo-fenchyle acetate (1.2–11.0), α-phellendrene (4.2–4.4), (Z)-β-ocimene (0.31–1.6), myrcene (0.8–1.3)	[21]
Black cumin seed oil	*Nigella sativa* L.	Seeds	(%) para-Cymene (37.3), thymoquinone (13.7), linalool (9.9), α-thujene (9.9), longifolene (6.4), β-pinene (3.4), α-pinene (3.1)	[22]
Caraway oil	*Carum carvi* L.	Seeds	(%) Carvone (76.8–80.5), limomene (13.1–16.2), γ-cadinene (0.30–0.46)	[23]
Cinnamon oil	*Cinnamomum* *Zeylanicum*	Barks	(%) (*E*)-Cinnamaldehyde (97.7), γ-codinene (0.9), α-copaene (0.8), α-amorphene (0.5)	[24]
		Leaves	(%) Eugenol (76.6–87.3), linalool (8.5), bicyclogermacrene (3.6), piperitone (3.3), eugenyl acetate (2.7), (Z) cinnamyl acetate (2.6), α-phellandrene (1.9), β-Caryophyllene (1.9)	[24,25]
Clove oil	*Eugenia Caryophyllata* (*Syzigium aromaticum* L.)	Buds	(%) Eugenol (88.6), eugenyl acetate (5.6), β-caryophyllene (1.4), 2-heptanone (0.93)	[26]
Coriander oil	*Coriandrum sativum* L.	Fruits	(%) Linalool (72.2–87.5), α-pinene (2.1–5.9), γ-terpinene (2.7–5.6), camphor (3.0–4.9), geraniol (1.9–3.9), geranyl acetate (0.8–2.9)	[27,28]
Dill oil	*Anethum graveolens* L.	Top plant	(%) Phellandrene (33.0–37.9), carvone (25.5–32.5), limomene (14.1–18.1), dill ether (3,9-epoxy-1-*P*-menthene; 7.5–10.8), α-pinene (0.85–1.15)	[29]
Garlic oil	*Allium sativum*	Bulb	(%) Diallyl disulfide (53.0), diallyl trisulfide (11.5), diallyl monosulfide (10.6), methyl allyl trisulfide (7.0), methyl allyl disulfide (4.4), diallyl tetrasulfide (4.3), methyl allyl tetrasulfide (2.5)	[30]
Laurel oil	*Laurus nobilis* L.	Leaves	(%) 1,8-Cineole (23.5), α-terpinyl acetate (10.8), linalool (10.6), methyl eugenol (9.4), sabinene (4.2), α-terpineol (3.9), terpin-4-ol (3.3), α-pinene (3.2), β-pinene (2.7)	[31]
Lavender oil	*Lavandula angustifolia*	Flowers	(%) Linalool (21.7–44.5), linalyl acetate (32.7–43.1), terpinen-4-ol (3.1–6.9), caryopyllene (5.0), 1,8- cineole (4.8), borneol (3.9), α-terpineol (3.5)	[32,33]
Lingonberry	*Vaccinium vitis- idaea*	Fruits	(µg/g fresh weight) Cyanidin 3-galactoside (486.9), quercetin 3-galactoside (86.1), quercetin 3-rhamnoside (82.3), caffeic acid (61.6), yaniding 3-arabinoside (62.7), β- coumaric acid (61.6), quercetin derivates (48.7), peonidin 3-glucoside (41.3), quercetin 3-arabinoside (29.9)	[34]
Lemon oil	*Citrus Limon*	Fruits	(%) Limonene (65.6–69.9), sabinene (11.2–13.0), γ- terpinene (1.9–2.1), myrcene (1.7), geranial (1.4–1.7), neral (0.8–1.0)	[35]
Mountain pride oil	*Heracleum persicum*	Fruits	(%) Hexyl butyrate (56.5), octyl acetate (16.5), hexyl 2-methylbutanoate (5.2), n-octanol (1.4), p-cymene (1.3), n-octyl 20methylbutyrate (1.5), n-hexyl hexanoate (1.3), n-hexyl butyrate (1.3)	[36]
Nutmeg oil	*Myristica fragaans*	Fruits	(%) α-pinene (22.2), sabinene (20.2), β-pinene (15.1), myristicin (9.6), terpinen-4-ol (4.2), and γ-terpinene (4.1), safrole (1.7)	[37]
Quebracho extract	*Schinopsis lorentzii*, *Schinopsis Balansae*	heartwoods	Catechin, *ent*-fisentinidol-4-ol	[38]
Oregano oil	*Origanum vulgare*	Aerial (Flowers, leaves)	(%) Thymol (63.3), γ-terpinene (12.7), *P*-Cymene (9.9), carvacrol (7.8), α-terpinene (1.0)	[32]
Sainfoin	*Onobrychis viciifolia*	Young leaves Young petiols	(mg/g DM) Rutin (19.9), isorhamnetin 3-O-rutinoside (3.56), nicotiflorin (2.82), quercetin 3-O-rhamnosylrutinoside (2.14), (mg/g DM) Arbutin (17.7), rutin (9.14) isorhamnetin 3-O-rutinoside (3.56), catechin (3.46), 8-β glucopyranosyloxycinnamic acid	[39]
Turmeric oil	*Curcuma longa* L.	Rhizomes	(%) 1,8-cineole (11.2), α-turmerone (11.1), β-caryophyllene (9.8), α-phellandrene (8.0), *ar*-turmerone (7.3), β-sesquiphellandrene (7.1), zingiberebe (5.6), β-turmerone (5.0), *ar*-curcumene (4.4), β-curcumene (4.2), caryophyllene oxide (3.4), β-bisabolene (2.8)	[25]
Thyme oil	*Thymus vulgaris*	Aerial (Leaves, flowers)	(%) Thymol (19.4–54.1), P-cymene (11.6–32.2), γ-terpinene (1.1–23.3), β-caryophyllene (2.0–5.3), carvacrol methyl ether (1.6–5.0), carvacrol (1.4–4.0), α-terpinene (0.6–3.5), linalool (0.7–2.2), 1,8-cineol (0.9–2.5), myrcene (0.2–2.3), α-thujene (0.15–2.9).	[40]
Wattle extract	*Acacia mearnsii*	Barks	(%) Robinetinidol–catechin–robinetinidol (32), robinetinidol–gallocatechin–robinetinidol (27), robinetinidol–catechin–fisetinidol (20), robinetinidol–gallocatechin–fisetinidol (13), fisetinidol–catechin–fisetinidol (5), fisetinidol–gallocatechin–fisetinidol (3)	[41]
Wattle	*Acacia mangium,* *Acacia auriculiformis*	Heartwoods	2,3-trans-3,4′,7,8-tetrahydroxyflavanone, teracidin, 4′,7,8-trihydroxyflavanone	[42]

Monoterpenes comprise several functional radical constituents, such as carbures, alcohols (i.e., menthol, geraniol, and limomene), aldehydes, ketones, esters, ethers, peroxide, and phenols, whilst sesquiterpenes have almost the same structure and role as monoterpenes, and broadly accumulate together with monoterpenes [6]. Diterpenes are acid components in the resins of gymnosperms, such as abeitic acid and other compounds, for example phytol, tocopherol, and retinol [6]. Chemical constituents of EO in each plant may vary depending upon the plant tissues, such as stems, leaves, fruits, flowers [43], genotypes, cultivars [27,44], maturity, environment, and regions [20,23,27].

Some examples of EO from aerial parts in the form of leaves and flowers include oregano oil (*Oreganum vulgare*) and thyme oil (*Thymus vulgaris*), which both contain thymol at proportions of 63.3% [32] and 19.5–54.1% [40], respectively. Other EOs derived from leaves and flowers are basil oil (*Ocimum basilicum* L.), with an estragole content of 52.6–58.3% [21], while dill oil (*Anethum graveolens* L.) contains 33–38% phellandrene [29]. EO can also be obtained from fruit parts, such as anise oil (*Pimpinella aisum* L.), coriander oil (*Coriandrum sativum* L.), lemon oil (*Citrum limon*), mountain pride oil (*Heracleum persicum*), and nutmeg oil (*Myristica fragaans*), with the main content of each EO in the form of trans-anethole, at 76.9–93.7% [20], linalool, at 72.2–87.5% [27,28], limonene, at 65.6–69.9% [35], hexyl butyrate, at 56.6% [36], and α-pinene, at 22.2% [37].

EOs can be obtained from not only the previously mentioned plant parts, but also other parts such as seeds, rhizomes, tree bark, tubers, and buds. *Nigella sativa* L. and *Carum carvi* L. are examples of medicinal plants where EOs are extracted from seeds as black cumin seed oil and caraway oil, respectively. The primary active compound in black cumin seed oil is para-cymene (37.3%) [22], while caraway oil contains carvone (76.8–80.5%) [23]. Other EOs such as turmeric oil, with 1,8-cineole (37.3%) being the main constituent, are taken from the rhizome *Curcuma longa* L. [25]. Cinnamon oil (E-cinnamaldehyde, 97.7%) is derived from the bark of the *Cinnamomum zeylanicum* tree [24], while clove oil (eugenol 88.6%) was extracted from the buds of *Eugenia caryophyllata* [26], and garlic oil (diallyl disulphide, 53%) was extracted from *Allium sativum* tubers.

#### 3.1.1. Effect of Essential Oils as Feed Additives for Ruminants

Table 2 reviews various research findings on the effects of EO, in the form of either extracts or whole plants, on ruminant fermentation profiles, gas (GP) and CH_4_ productions, and animal performance and health. Patra and Yu [45] reported that various EO supplementations reduced degradability, GP, and CH_4_ output, in line with decreasing archaea, protozoa, and cellulolytic bacteria. Protozoa and the majority of cellulolytic bacteria produce H_2_ as their end product of fermentation, which is mainly utilized by methanogens (archaea) to form CH_4_ in the rumen [46,47]. Lower CH_4_ can be produced where more H_2_ can be competitively converted, along with carbon dioxide (CO_2_), to form acetate by hydrogenotrophic acetogens [48,49]. However, acetogens are able to utilize H_2_ and CO_2_ to produce acetate in the rumen, where methanogens are greatly inhibited [50]. If acetogenesis is dominant over methanogenesis, it can result in the predominant uses of H_2_ and CO_2_ by acetogens to produce acetate [48,51]. Reduced rumen CH_4_ formation due to EO supplementations were also reported by other investigations [52,53].

**Table 2 animals-13-00767-t002:** Effects of different essential oils as dietary additives for various ruminant animals.

No.	Essential Oils	Basal Control Diets	Test Systems	Outputs	References
1.	Clove oil (CLO), eucalyptus oil (EUC), garlic oil (GAR), origanum oil (ORI), and peppermint oil (PEP) at 0.25, 0.50, and 1.0 g/L in vitro fermentation medium	Ground alfalfa and dairy concentrate mix (50:50)	In vitro dairy cows	Increasing doses of all EO reduced tGP up to 79.4% and CH_4_ up to 86.9% at 1 g/L but reduced IVDMD except GAR; reduced NH_3_ for CLO and ORI; increased pH; increased VFA for EUC, GAR, and PEP but reduced VFA for ORI; increased A:P ratio for CLO, ORI, and PEP but decreased A:P for EUC and GAR; increased butyrate; decreased archaea, protozoa, and major cellulolytic bacteria	[45]
2.	Experiment 1: Ground cinnamon bark (CIN), clove buds (CLO), coriander seeds (COR), cumin seeds (CUM), turmeric roots (TUR) Experiment 2: COR, CUM, TUR, and combination between COR, CUM, and TUR (MIX) (30 mg/g substrate)	Exp. 1: wheat-based mixture substrate Exp. 2: Ryegrass hay-based mixture substrate	In vitro Sheep	Exp. 1: no effect on IVDMD except being lower for CIN; no effect on pH; increased NH_3_ for COR and CUM; increased tVFA except for COR and TUR; decreased acetate for CLO and COR but no effect on A:P; decreased CH_4_ by 21.5–44.8% except for CIN Exp. 2: no effect on IVDMD except being lower for MIX; no effect on pH; decreased NH_3_ except for CUM; no effect on tVFA but decreased A:P for COR and CUM; decreased CH_4_ by 22.0–67.0% for all spices additions	[53]
3.	Garlic oil (GAR), cinnamon oil (CIN), thyme oil (THY), coriander oil (COR), caraway oil (CAR), cumin oil (CUM), nutmeg oil (NUT), dill oil (DIL), rosemary oil (ROS), red basil oil (RBA), oregano majorana oil (ORM), oregano vulgare oil (ORV), mountain pride oil (MOP), clove oil (CLO), lemon oil (LEM), black pepper oil (BLP), fennel oil (FEN), Peppermint oil (PEP), and pistachio oil (PIS) at 1 µL/50 mL rumen-buffered fluid each	Ground alfalfa hay and concentrate (80:20)	In vitro Sheep	Almost all the EO decreased tGP by 25.2–95.5% except for FEN, BLP, PEP, ROS, PIS, DIL, CLO; decreased IVDMD and IVCPD except for BLP, ROS, DIL; increased pH but decreased pH for only BLP, ROS, DIL and no effect for FEN, ORM, CIN, GAR; decreased NH_3_ except for FEN and MOP; decreased CH_4_ for COR, CIN, REB, ORV, CUM, CAR, DIL by 11.6–76.7% but no effect for ROS and BLP	[52]
4.	Oregano vulgare (ORV), black seed (BLS), laurel (LAU), cumin (CUM), garlic (GAR), and anise (ANI), cinnamon (CIN) at 50, 100, and 150 ppm	Either barley, SBM, or wheat straws	In vitro dairy cows	Across incubation hours, all doses of CUM increased tGP, while ORV at 100 or 150 ppm decreased tGP in all substrate basal diets; GAR (150 ppm) decreased tGP in barley and wheat straw-based diet; ANI (almost all doses) decreased tGP in all substrates	[54]
5.	About 400 mg blended EO (266 mg Cinnamaldehyde (CIN) and eugenol (EUG) + 133 mg capsium oleoresin (CAO) per steer in a mineral mixture with Monensin (46.7 mg/kg DM) as a control	Corn grain-based concentrate (ad- libitum) + 200 g as fed alfalfa/steer/d	In vivo feedlot cattle	No effect on DMI, FCR, and VFA profiles but decreased NH_3_ (0–84 d); increased ADG between 45 and 84 d	[55]
6.	A mixture EO consisting of thymol, eugenol, vanillin, and guaiacol, limonene (Crina Ruminants, Switzerland) at 50, 100, and 150 mg/kg DM of concentrate	Lucerne hay and dairy concentrate mix (50:50)	In vivo dairy ewes	Increased milk production (L/ewe/d), from 1.57 (control) to 1.68, 1.88, and 2.12 (50, 100, 150 mg EO/kg, respectively) but no effect on milk composition, as well as reduced urea concentration and somatic cell count at the greatest dose; no effect on cellulolytic bacteria and protozoa but decreased hyper-NH_3_-producing bacteria; no effect on pH; reduced NH_3_ and increased tVFA at the highest dose; decreased A:P	[56]
7.	CE Lo (0.5 g/d, 85 mg Cinnamaldehyde + 140 mg eugenol), CE Hi (10 g/d, 1700 mg Cinnamaldehyde + 2800 mg eugenol), CAP (0.25 g/d, 50 mg Capsium)	Forage and dairy concentrate mixture (48:52) (DM basis)	In vivo dairy cows	No effect on DMI, VFA, A:P, NH_3_, milk yield, fat and protein in milk (kg/d), NDF and ADF disappearances reduced with CE Hi	[57]
8.	A mixture of EO (7% eucalyptus oil, 6.6% menthol crystal, 2% mint, 22.5% ethanol, 15.3% emulsifiers, and demineralized water up to 100%, Kanters Special Product Co, Netherland) at 16, 32, and 48 mg/L of drinking water	Berseem hay and dairy concentrate mixture (50:50)	In vivo dairy cows	No effect on feed intake, Increased water intake for dose 48 mg/L; no effect on DM, OM, CP digestibility, milk production, and fat but increased protein in milk; no effect on pH and NH_3_ but increased VFA for doses 16 and 32 mg/L; decreased A:P for 16 and 32 mg/L but increased A:P for 48 mg/L; no effect on total viable bacteria, cellulolytic, and protozoa counts for all doses of EO	[58]
9.	Cinnamaldehyde (CIN) (>98% purity), garlic oil (GAR) (1.5% allicin), or Juniper berry (JUN) (35% α-pinene) (Pancosma S.A., France) at 200 mg/kg DM of diet	Barley-based concentrate and alfalfa hay (84:16)	In vivo lambs	No effect on DMI but CIN and JUN had higher ADG and less blood glycerol than GAR and the control; No different on pH, NH_3_, VFA, nor A:P; only CIN had higher total blood triglycerides; all additives gave higher liver weight than the control but no difference for hot dressed weight, weight of cuts, and saleable meat yield; all additives had minor effects on the overall fatty acid compositions (back fat and liver) and meat flavor characteristics	[59]
10.	Cinnamaldehyde (CIN) (>99% purity) and carvacrol (CAR) (>98% purity) (Phodé S.A., France) at 200 mg/kg DM diet	Either barley-based or corn-based diets	In vivo lambs	No difference for DMI, ADG, and NH_3_; CIN and CAR increased tVFA in both barley- and corn-based diets but no difference in A:P; no difference for carcass characteristics, meat yield, and sensory evaluations	[60]
11.	Oregano oil (carvacrol 83.1%, thymol 2.1%, γ-terpinene 4.0%, p-cymene 3.8%, and β-caryophyllene 0.9%) at 1 mL/kg diet	Maize-based diet and alfalfa hay (55:45)	In vivo lambs	No effect on DMI, ADG, hot carcass weight, carcass yield, and tenderness; increased pH and color of meat; decreased lipid oxidation during refrigerated and long-term frozen storage	[61]
12.	Thyme (thymol, carvacrol, P-cymene, γ-cadinene) 20 g/ewe/day + basal diet. Celery (limonene, γ-cadinene, thymol) 20 g/ewe/day + basal diet	Concentrate: fodder maize (*Zea mays* L.) 60:40	In vivo lactating ewes	Thyme and celery increased weight gain and milk production. Thyme enhanced feed intake and nutrient digestibility. Thyme is preferred to celery in the diets of lactating ewes.	[62]

The use of EO as a dietary additive for ruminants in this article focuses on the research conducted in vivo. The in vitro research was still included, since it has been continued with the in vivo tests. It is intended that the utilization of EO as a dietary additive has confirmed its effect on livestock directly. Several parameters discussed in this article are related to the effect of EO on in vivo ruminant performances, for example, feed intake, body weight gain, feed efficiency, and nutrient digestibility. If the research was preceded by an in vitro test, the parameters to be observed were in vitro dry matter (IVDMD) and organic matter (IVOMD) digestibility, volatile fatty acids (VFA), ammonia (NH_3_), and CH_4_ outputs.

An in vivo experiment to observe the effect of EO on ruminants was carried out to investigate the use of a more mixed form of EOs compared to a single form. Geraci et al. [55] investigated a mixture of cinnamaldehyde and eugenol with a total administration of 400 mg/steer mixed into the mineral mixture. The same mixture with different concentrations was also tested by Tager and Krause [57] in dairy cows. The EO blend used by Geraci et al. [55] and Tager and Krause [57] reported no effect on DMI and VFA profiles in both feedlot cattle and dairy cows, respectively. A mixture of EO consisting of thymol, eugenol, vanillin, guaiacol, and limonene (Crina Ruminants, Switzerland) at 50, 100, and 150 mg/kg DM, respectively, given to dairy ewes, showed an improvement in milk production, but it had no impact on the milk compositions [56]. Soltan et al. [58] also reported insignificant effects of EO supplements containing eucalyptus oil, menthol, and mint on feed intake, DM, OM, CP digestibility, and milk production, except for enhanced milk protein in dairy cows.

Chaves et al. [59] observed that cinnamaldehyde and juniper berry EO additions in the diet increased average daily gain (ADG) but other studies reported that cinnamaldehyde, carvacrol [60], and oregano [61] EO supplementations had no effect on ADG in growing lambs.

It was reported that EO additions in the diet of growing lambs had no impact on carcass weight, meat yield [59,60,61], sensory parameters [60], tenderness [61], meat flavor, or overall fatty acid compositions [59]. However, Simitzis et al. [61] observed increased pH and the color of meat lambs as the result of EO supplementation, and a decrease in lipid oxidation during refrigeration and long-term freezing.

Research using EO mixture showed that the obtained effect was not considerably significant, and it was difficult to define which EO had the strongest influence. By using the same EO mixture, the effect of different inclusion levels can also have different impacts on livestock. Thus, this needs to be studied more deeply by considering the role of each EO containing different chemical substances.

It seems that the use of EOs as dietary additives to mitigate CH_4_ output by the rumen in in vitro evaluations is nearly conclusive. However, the results of the effects of various EO inclusions into different ruminant diets on GP, VFA profiles, NH_3_, pH, and feed degradability parameters are still inconsistent. This is understandable, since there are naturally many sources of EO, and each of them may have different chemical constituents, so that the interaction among the chemical components of EO, doses, nutrient characteristics of different diets, and existing microbial populations in the rumen needs to be appropriately understood when planning similar research in the future.

#### 3.1.2. Effect of Essential Oils as Anthelmintics for Ruminants

Table 3 summarizes the results of several studies using EO as anthelmintics for ruminants. The EO supplementation is also beneficial in improving animal health by combating parasites. Adding both *Eucalyptus staigeriana* [63] and *Lippia sidoides* [64] EOs in the diets of goats and sheep, respectively, was effective in helping animals against gastrointestinal nematodes, such as *Haemonchus* spp. and *Trichostrongylus* spp.

**Table 3 animals-13-00767-t003:** Main outcomes of many studies that used essential oils as anthelmintics for ruminants.

No	Essential Oils	Test Systems	Outputs	References
1.	*Eucalyptus staigeriana* oil (Dierberguer óleos essenciais Ltd., Brazil) at 1.35 and 5.4 mg/ml	In vivo Sheep infected with *Haemonchus contortus*	Both doses reduced fecal egg hatching and larval development of *Haemonchus contortus* by 99.3 and 99.2%, respectively. The efficacy of the EO against gastrointestinal nematodes was 76.6% at 15th day after treatment	[63]
2.	*Lippia sidoides* oil (LIP) (Pronat, Brazil) at 230 and 283 mg/kg animal. Positive control: Ivermectin at 200 µg/kg	In vivo naturally infected sheep	Increased the efficacy against gastrointestinal nematodes by 38% (230 mg/kg), 45.9% (283 mg/kg), and 40.2% (Ivermectin) 7 days after treatment and 30%, 54% and 39.6%, respectively, 14 days after treatment LIP oil (283 mg/kg); Ivermectin increased the respective efficacy by 56.9% and 34.4% against *Haemonchus* spp. and 39.3% and 63.6% against *Trichostrongylus* spp.	[64]
3.	Flaxseed oil (3%) for the first and Vitamin E (0.06 g/kg DM) for the second sheep groups. Control: sheep without additives	In vivo infected sheep by *Fasciola hepatica*	Flaxseed oil supplementation showed a reduction in adult fluke burden, smaller flukes recovered at necropsy, and a lower fecal egg count at the end of trial. Vitamin E led to reduction in adult fluke burden and lower lipid oxidation in the liver	[65]
4.	*Artemisia lancea oil at* 10, 5.0, 2.5, 1.25, 0.63 mg/ml	In vitro anthelmintic assays using the eggs or adult nematodes from faeces of parasite-infected donor sheep	The essential oil of *Artemisia lancea* had an anthelmintic activity against eggs and larvae of *Haemonchus contortus*	[66]
6.	*Eucalyptus citriodora* (citronellal 63.9%), citronellal (purchased from Sigma-Aldrich^®^) at different concentrations: 1. egg hatch test (EHT, 0.125–2 mg/mL), 2. Larval development test (LDT, 0.5–8 mg/mL), 3. Adult worm motility test (AWMT, 1–2 mg/mL). Control: 1. EHT ((−) control 1% Tween^®^80, (+) control 0.025 mg/mL thiabendazole), 2. LDT ((−) control 1% Tween^®^80, (+) control 0.008 mg ivermectin/mL), 3. AWMT ((−) control 4% penicillin/streptomycin, (+) control 100 μg ivermectin/mL)	In vitro tests of EHT, LDT, AWMT from two infected sheep with 5000 *Haemonchus contortus* L_3_, the inbred-susceptible Edinburgh (ISE) isolate, and the other with 5000 *Haemonchus. contortus* L3, the Kokstad isolate	EHT (EC50 value): ISE Isolates were 0.4 mg/mL of *Eucalyptus citriodora* oil and 0.3 mg/mL of citronellal. The Kokstad isolates were 0.5 mg/mL of *Eucalyptus citriodora* oil and 0.4 mg/mL of citronellal. In AWMT, 2 mg/mL of oil and citronellal completely inhibited the motility of both the ISE isolate and Kokstad isolate. Both *Eucalyptus citriodora* oil and citronellal exhibited ovicidal and larvicidal effects and inhibited the motility of both *Haemonchus contortus* isolates	[67]
7.	*Artemisia campestris* aerial parts: 2, 4, or 5 g EO/kg (-) control: 3% tween^®^80 (+) control: 22 mg albendazole/kg	In vitro assays: Anthelmintic activity test on *Haemonchus contortus* using egg-hatching assay (EHA) and adult worm motility assay (AWMA)*;* Nematocidal effect test on *Haemonchus polygyrus* with FECR (fecal egg count reduction) and TWCR (total worm count reduction)	The dominated EOssin *Artemisia campestris* were beta-pinene (36.40%) and 2-undecanone (14.7%) EHA: 100% inhibition was observed at 2 mg/mL after 48 h incubation. AWMA: 66.6% inhibition at 0.5 mg/mL after 8 h post exposure The dose of 5 g/kg EO showed a high nematocidal activity (72.1% FECR and 72% TWCR)	[68]
8.	*Thymus vulgaris* EO (300, 150, and 75 mg/kg body weight, BW) and Monoterpene thymol. (+) control: 2.5 mL/kg BW of Zolvix^®^, (−) control: 1 mL/kg BW of saline.	In vivo infected sheep with 4000 L_3_ larvae of a resistant isolate*,* in vitro anthelmintic assay and in vivo sheep	*Thymus vulgaris* EO and thymol could inhibit egg hatching (*Haemonchus contortus*) by 96.4 to 100%, larval development by 90.8 to 100%, and larval motility by 97 to 100%	[69]
9.	*Zanthoxylum simulans* EO (ZSEO), borneol, and β-elemene at 40, 20, 10, 5, 2.5, and 1.25 mg/mL. (+) control: albendazole (EHA, LDA), levamisole (LMIA). (−) control: Phosphate-buffered saline and tween^®^80	In vitro anthelmintic assay using the EHA, larval development assay (LDA), and larval migration inhibition assay (LMIA), with sheep infected by 10,000 *Haemonchus contortus L_3_*	ZSEO (40 mg/mL) inhibited larval hatching by 100% with LC_50_ values of 3.98 and 1.50 for borneol. LDA results showed that ZSEO, borneol, β-elemene at 40 mg/mL inhibited larval development by 99.8%, 100%, and 55.4%, respectively. LMIA showed that ZSEO, borneol, and β-elemene inhibited larval migration by 74.3%, 97.0%, and 53.2%, respectively	[70]
10.	*Citrus sinensis* and *Melaleuca quinquenervia* EO at 0.02–50 mg/mL (EHT) and 0.04–3.12 mg/mL (LDT). (+) control: thiabendazole (EHT), ivermectin (LDT) (-) control: Tween^®^80 (EHT), 05% DMSO (LDT)	In vitro assays using EHT and LDT with two sheep infected by *Haemonchus. Contortus,* fed by 400 g of corn and silage	*Citrus sinensis* contained limonene as a major component (96.0%), *Melaleuca quinquenervia* contained longifolene (32.95%) and 1,8-cineole (25.43%) as major components. EHT: IC50 and IC90 of the EO were 0.27 and 0.99 mg/mL for *Citrus sinensis* and 1.52 and 5.63 mg/mL for *Melaleuca quinquenervia,* respectively LDT: IC50 and IC90 of the EO were 0.97 and 2.32 mg/mL for *Citrus sinensis* and 0.44 and 0.94 mg/mL for *Melaleuca quinquenervia*, respectively. *Citrus sinensis* was more effective on eggs but *Melaleuca quinquenervia* was twice more effective on larvae.	[71]
11.	*Ruta chalapensis* leaves and flower EO at 0.05, 0.1, and 0.05% for insecticidal activity evaluation and at 0.125, 0.25, 0.5, and 1% for in vitro anthelmintic assay. Control of insectisidal activity: 0.015% Decis (+) and 96% ethanol (−). Control in AWMA: 1 mg/mL albendazole (+), PBS (−)	Insecticidal activity evaluation with *Orgyia trigotephras* larvae fed on *Erica multiflora* fresh leaves, as well as in vitro anthelmintic assays conducted using the EHA and AWMA, with *Haemonchus contortus* from the feces and abomasum of experimentally infected lambs	*Ruta chalepensis* EO from flowers and leaves showed significant insecticidal and anthelmintic activites	[72]
12.	*Mentha piperita, Cymbopogon martini, Cymbopogon schoenanthus* EO at 2%.	In vitro assay using EHA, LDA, LFIA (larval feeding inhibition assay), and LEA (larval ex-sheathment assay) with sheep naturally infected by 95% *Haemonchus. contortus* and 5% *Trichostrogylus* spp.	The major constituent of the EO for *Mentha piperita* was menthol (42.5%)*,* for *Cymbopogon martini* it was geraniol (81.4%)*,* and for *Cymbopogon schoenanthus* it was geraniol (62.5%). *Cymbopogon schoenanthus* EO had the best activity against *Ovine trichostrongylids* followed by *Cymbopogon martini,* while *Mentha piperita* showed the least activity.	[73]

Research on the effect of EO on reducing parasites and improving ruminant health was also carried out using an in vitro method. These in vitro experiments have been done to examine the presence of anthelmintic activities of various types of EO. The researchers have used different parasites in various growth phases such as eggs, larvae, and adult parasites using different methods of assessments. Most of the EO treatments indicated a reduction in the number of eggs and larvae of *Haemonchus contortus* [66,68,74]. Ferreira et al. [69] concluded that EOs from *Thymus vulgaris* could inhibit egg hatching, as well as the larval development and motility, of *Haemonchus contortus* in sheep. 

*Haemonchus contortus, Trichostrongylus* spp., *Fasciola hepatica*, *Rhipicephalus microplus*, and *Haemonchus polygyrus* are widely studied as harmful parasites to ruminants. Several studies took EOs from various types of plant parts, especially those above the ground (not roots). Macedo et al. [63] conducted a study using EOs derived from *Eucalyptus staigeriana* in sheep infected with *Haemonchus contortus*. The results showed that EOs from *Eucalyptus staigeriana* was able to reduce worm eggs and larval development, and combat nematodes in the digestive tract of sheep. Similarly, a study performed by Camurça-Vasconcelos et al. [64] confirmed that EOs from *Lippia sidoides* increased the ability to combat nematodes such as *Haemonchus contortus* and *Trichostrongylus* spp. in sheep. Additionally, the inclusion of about 3% flaxseed oil in the diet of sheep could reduce the number of fecal egg counts [65]. 

An anthelmintic effect was also shown by EOs derived from the flowers and leaves of *Ruta chalapensis* [72]. The EO was tested in vitro using *Haemonchus contortus* derived from goats, and compared with albendazole. The results showed that EOs from leaves gave a higher inhibitory impact on worm hatching than EOs from flowers. Meanwhile, EOs derived from flowers showed an inhibition of motility of up to 87.5% after 8 h of exposure.

As mentioned earlier, the anthelmintic test of medicinal plants can use various types of methods. A study conducted by Katiki et al. [73] evaluated the anthelmintic *Cymbopogon schoenanthus* against *Trichostrogylus* spp. by using different methods, namely, the egg hatch assay (EHA), larval development assay (LDA), larval feeding inhibition assay (LFIA), and larval ex-sheathment assay (LEA). All of these methods validated that *Cymbopogon schoenanthus* potential as an anthelmintic, although it had to be retested in vivo. Similarly, *Zanthoxylum simulans’* EO has been tested in vitro using the EHA, LDA, and larval migration inhibition assay (LMIA), which confirmed that this EO had an anthelmintic potential to inhibit larval development of *Haemonchus contortus* in sheep [70].

The effects of EOs as anthelmintics are related to the interaction of these compounds with the structure of the parasite. This occurs when the lipophilic compounds, such as essential oil constituents, can break or damage the cell membrane of the parasite, thus affecting membrane permeability and leading to some enzyme and nutrient losses [74]. It is also possible that these Eos inhibit cell growth and differentiation, a very rapid process of worm egg embryogenesis [71].

### 3.2. Polyphenol Sources, Types and Uses

Polyphenols, such as tannins, are plant bioactive substances with various molecular weights and complexities. These compounds can bind to dietary proteins in aqueous solutions [75,76]. Although some pure plant polyphenols may be rarely soluble in water, their natural interactions ensure that some of those can be soluble in aqueous media [77]. Tannins contain multiple phenolic hydroxyl units that are able to configure complexes majorly with proteins, and minorly with metal ions, amino acids and polysaccharides [75]. Broadly, tannins are divided into two major groups: hydrolysable and condensed tannins (CT).

Hydrolysable tannins, known as gallotannins and ellagitannins, contain a structure based on a gallic acid unit. These are commonly identified as polyesters with D-glucose (gallotannins), while derivatives of hydroxydiphenic acid (ellagitannins) are developed from the oxidative coupling of contiguous gallolyl ester groups in a polygallolyl D-glucose ester [78]. Haslam [78] illustrated two pathways of gallic acid biosynthesis: (a) direct dehydrogenation of an intermediate in the shikimate pathway, as well as the retention of oxygen atoms of the alicyclic precursor, (b) a derivative of the end-product of the pathways. 

Condensed tannins (CT), or proanthocyanidins, are structured by a nucleophilic flavanyl group, often a flavan-3-ol (‘catechin’) that is generated from an electrophilic flavanyl unit, flavan-4-ol, or flavan-3,4-diol [16]. Proanthocyanidins occur as water-soluble oligomers comprising two, to ten or more, ‘catechin’ groups and water-insoluble polymers [78]. Due to differences in hydroxylation patterns, Bruyne et al. [16] have classified proanthocyanidins into a number of subgroups: propelargonidins (3,4′,5,7-OH), procyanidins (3,3′,7-OH), prodelphinidins (3,3′,4′,5,5′,7-OH), proguibourtinidins (3,4′,7-OH), profisetinidins (3,3′,4′,7-OH), prorobinetinidins (3,3′,4′,5′,7-OH), proteracacidins (4′,7,8-OH; only synthetic), promelacacidins (3′,4′,7,8-OH), proapigennidins (4′,5,7-OH), and proluteolinidins (3′,4′,5,7-OH). They reported that procyanidins mostly appear in barks or woody plants, and were the commonest, whilst the prodelphinidins were the main substances of the leaves and conifers. 

Tannins contained in plants can be found in all parts of the plant, such as in sainfoin (*Onobrychis viciifolia*), with the largest content of Quercetin 3-rutinoside (6.15 mg/g DM) [79]. In addition to the plant as a whole, tannins are also found in leaves, young leaves, tree stalks, tree bark, core wood, and fruits. Leaves of *Camellia sinensis* (green tea), *Pistachia lentiscus*, and *Phillyrea latifolia* are known to contain tannins, where their dominant tannin contents are epigallocatechin gallate (94.6 mg/g DM) [7], cholorogenic acid (17.4 mg/L), and oleuropein (167.0 mg/L) [80], respectively. Several other plants containing tannins are described in Table 4.

**Table 4 animals-13-00767-t004:** Major bioactive compounds in different parts of some polyphenol-rich plants.

Plants	Scientific Names	Main Parts	Major Bioactive Compounds	References
Green tea	*Camellia sinensis*	Leaves	(mg/g DM) Gallocatechin (4.93), epigallocatechin (22.4), catechin (1.30), epicatechin (2.13), epigallocatechin gallate (94.6), gallocatechin gallate (1.15), epicatechin gallate (25.5), catechin gallate (3.10), theaflavin (0.28), theaflavin-3-gallate (0.22), theaflavin-3′-gallate (0.35), theaflavin-3,3′-digallate (0.38)	[7]
Lingonberry	*Vaccinium vitis- idaea*	Fruits	(µg/g fresh weight) Cyanidin 3-galactoside (486.9), quercetin 3-galactoside (86.1), quercetin 3-rhamnoside (82.3), caffeic acid (61.6), cyanidin 3-arabinoside (62.7), β- coumaric acid (61.6), quercetin derivates (48.7), peonidin 3-glucoside (41.3), quercetin 3-arabinoside (29.9)	[34]
Pistachio	*Pistachia lentiscus*	Leaves	(mg/L) Chlorogenic acid (17.4), 3,4,5 tri-O- galloyquinic acid (15.9), rutin (13.6), 3,5 di- O-galloyquinic acid (10.8), myricetin-3-O- rutinoside (6.8), catechin (5.6)	[80]
Zaitun	*Phillyrea latifolia*		(mg/L) Oleuropein (167.0), tyrosol (78.2), quercetin-7-O-rutinoside (42.5), apigenin-7-O-glucoside (20.0), quercetin (14.7), luteolin- 7-O-glucoside (8.6), luteoline (7.6)	[80]
Quebracho extract	*Schinopsis lorentzii*, *Schinopsis Balansae*	heartwoods	Catechin, *ent*-fisentinidol-4-ol	[38]
Sainfoin	*Onobrychis viciifolia*	Whole plant (bud stage)	(mg/g DM) Quercetin 3-rutinoside (6.15), arbutin (2.69), kaempferol 3-rutinoside (1.87), quercetin 3-rhamnosylrutinoside (1.00), isorhamnetin 3-rutinoside (0.38); 3′-caffeoylquinic acid (0.33), kaempferol 3-rhamnosylrutinoside (0.29), 5′-caffeoylquinic acid (0.28), epicatechin (0.26)	[79]
		Young leaves Young petiols Flower buds	(mg/g DM) Rutin (19.9), isorhamnetin 3-O- rutinoside (3.56), nicotiflorin (2.82), quercetin 3-O-rhamnosylrutinoside (2.14), (mg/g DM) Arbutin (17.7), rutin (9.14), isorhamnetin 3-O-rutinoside (3.56), catechin (3.46), 8-β-glucopyranosyloxycinnamic acid (1.94), quercetin 3-O- rhamnosylrutinoside (1.52), epicatechin (1.23) (mg/g DM) Rutin (5.78), nicotiflorin (1.31)	[39]
Wattle extract	*Acacia mearnsii*	Barks	(% from extract) Robinetinidol–catechin– robinetinidol (32), robinetinidol–gallocatechin–robinetinidol (27), robinetinidol–catechin–fisetinidol (20), robinetinidol–gallocatechin–fisetinidol (13), fisetinidol–catechin–fisetinidol (5), fisetinidol–gallocatechin–fisetinidol (3)	[41]
Wattle	*Acacia mangium*, *Acacia auriculiformis*	Heartwood	2,3-trans-3,4′,7,8-tetrahydroxyfl vanone, teracidin, 4′,7,8-trihydroxyflavanone.	[42]

#### 3.2.1. Effect of Tannins as Feed Additives on Ruminants

Tannins reduce the solubility and rumen degradability of most dietary proteins, due to their ability to bind proteins. As a consequence, they may decrease the rumen NH_3_ output and enhance the protein availability and non-NH_3_–N supply to be absorbed in the small intestine [6,14,76]. Even though NH_3_ is a main source of N for rumen microbes, its fast or over production can exceed the ability of microbes to use it. This may result in an excessive NH_3_ supply that, after absorption via rumen wall, can enter the blood stream, liver, and finally be excreted in urine as an N waste, causing potential risks for the environment [81,82].

**Table 5 animals-13-00767-t005:** Effect of tannins as feed additives on different ruminant animals by using the in vitro, in sacco, and in vivo methods.

No	Tannins	Basal Diets	Test Systems	Outputs	References
1.	*Chrysanthemun coronarium* at 20 mg/0.4 g control substrate	Concentrate:grass hay (70:30)	In vitro sheep	Increased tVFA and slightly increased acetate but decreased propionate	[83]
2.	Whole purple prairie clover (legume, *Dalea purpurea* vent.) at either vegetative (VEG) or flowering (FLO) stages	VEG contained (g/kg DM) 916 OM, 167 CP, 334, NDF and 58.6 CT; FLO had 935 OM, 134 CP, 482 NDF, and 94.0 CT	In vitro dairy cows	VEG had higher DM and NDF digestibility and N in residue than FLO; no difference for VFA profiles and NH_3_	[84]
3.	CT extract (*Leucaena leucephala)* at 20, 30, 40, and 50 g/kg DM	*Panicum maximum*	In vitro cattle	Reduced tGP, CH_4_ (40 g/kg DM, the lowest), and IVDMD (only for 50 g/kg DM); no difference in pH	[85]
4.	Sainfoin hay (SH, *Onobrychis viciifolia* Scop.) at 4 different growth rates with CT contents 63.5–114 mg/g DM	Alfalfa hay (AH) as low-tannins counterpart	In vitro cows	SH had higher OM digestibility, tGP, CH_4_, tVFA, and acetate but lower NH_3_ than AH; no different on propionate and A:P	[86]
5.	Sainfoin (*Onobrychis viciifolia* Scop.), representing different CT contents of 48.4–78.5 g/kg DM	Concentrate, hay, and corn silage (30:35:35)	In sacco dairy cows	Reduced DM and CP degradability at increased CT contents	[87]
6.	Either *Acacia pennatula* or *Enterolobium cyclocarpum* (ground pods) at 45% of each diet (iso- protein and energy)	Sorghum-based concentrate and hay (*B. brizantha*) (95:5)	In vivo sheep	Increased DMI, especially with *A. pennatula,* but decreased DM and OM digestibility; no effect on feed efficiency from hexose to calculated VFA and CH_4_	[88]
7.	Tannins extract (bark of *Acacia mearnsii*, Mimosa Central Cooperative Ltd., South Africa) at 163 g/d (TAN-1) and 326 g/d (TAN-2), or 0.9 and 1.8% CT DMI, respectively	Ryegrass supplemented with cracked triticale grain at 4.5 kg DM/cow/d	In vivo dairy cows	Reduced CH_4_ by 14–29% but decreased DMI and milk yield (especially in TAN-2); TAN-2 decreased fat (19%) and protein (7%) contents in the milk; no effect on protein and lactose contents; decreased digestible energy and N lost in urine	[89]
8.	Sericea lespedeza (SER, *Lespedeza cuneata*), either fresh (20.2% CT) or hay (15.3% CT) forms	Alfalfa (ALF), sorghum–Sudan grass (GRASS) (both low in CT, ≥0.03%)	In vivo goats	Fresh forages: SER had higher DM and GE intakes but lower DM digestibility, CH_4_, and ciliate protozoa than ALF and GRASS; SER had a higher N intake than GRASS but was lower than ALF; No difference for BW, ruminal pH, NH_3_, bacteria, and cellulolytic bacteria. Hay forages: SER had higher DM and GE intakes but lower DM and N digestibility, CH_4_, and ciliate protozoa than ALF and GRASS; SER had higher N intakes and pH than GRASS but similar intakes to ALF. SER had lower NH_3_ than ALF but similar levels to GRASS; no difference for BW, bacteria and cellulolytic bacteria counts	[90]
9.	Quebracho tannins extract (45.6% tannins, *Schinopsis lorentzii,* Figli di Guido Lapi S.pA, Italy) at 95.7–104 g/kg diet (DM basis)	Barley-based concentrate	In vivo lambs	Increased vaccenic acid (VA, C_18:1_ t11) but no effect on stearic acid (SA, C_18:0_) compositions in rumen fluid; Lowered SA/VA ratio; decreased *Butyrivibrio proteoclasticus*, *Butyvibrio fibrisolvens,* and protozoa*;* increased rumenic acid (cis-9, trans-11 CLA) (2-fold) and PUFA but reduced SFA from longissimus muscle	[91,92]
10.	Quebracho tannins extract (*Aspidosperma quebracho*, Tannin Co., Peabody, MA, USA) at 80 g/kg diet	Beet-pulp-based diet containing alkaloids, either gramine at 2 g/kg diet or methoxy-N,N-dimethyltryptamine at 0.03 g/kg diet	In vivo lambs	No effect on total DMI; total digested DM, energy or NDF but increased N digestibility, retained N, and digested N	[93]
11.	Quebracho tannins (Unitan SAICA, Chaco, Argentina) (11%) + wheat bran (89%) at 400–500 g to obtain 4% tannins in the diet	Either a high-degradable protein diet (HP) (22% CP and 17% RDP) or low- degradable protein diet (LP) (11% CP and 8% RDP)	In vivo Wethers	Minor effect on intakes, although it tended to decrease intakes in HP diet; decreased NH_3_ and blood–urea N, especially in HP diet.	[94]
12.	Tannins extract (*Vaccinium vitis idaea*, Herbapol Poznan, Poland) at 140 g or 2 g tannins/kg diet DM	Lucerne, corn silages, meadow hay, and concentrate (forages:concentrate, 60:40)	In vivo dairy cows	Decreased pH, NH_3_, calculated CH_4_, protozoa; no effect on tVFA but reduced A:P; no effect on milk yield, fats, CP, lactose, and energy contents in milk, DM, OM, and NDF digestibility	[95]
13.	Green tea dust (*camellia sinensis*, 25.6 phenols, 23.0 tannins) at 0, 5, 1, 1.5, and 2% concentrations	Paddy straw hay:concentrate (30:70)	In vivo lambs	Increased ADG without any harmful impact on feed intake and nutrient digestibility	[14]
14.	Pine bark (3.2% condensed tannin DM in diet; treatment 30% pine bark + concentrate)	Bermudagrass hay+ concentrate (30:70)	In vivo male kids	The 30% pine bark supplementation did not show a negative effect on animal performance, blood metabolites, orand carcass parameters	[96]
15.	*Acacia mearnsii* extract (700 g/kg CT). Treatment: 0, 20, 40, 60, and 80 g CT/kg total DM diet	TMR with roughage:concentrate 40:60.	In vivo lambs	Recommendation of using *Accasia mearnsii* in lamb diet up to 40 g CT/kg DM, due to increased nutrient intake, digestibility, growth performance and feed efficiency.	[97]

The impacts of tannins as natural additives in various diets of ruminant have been studied using different in vivo, in vitro, and in sacco methods. Guglielmelli et al. [86] found that adding Sainfoin hay into a diet of cows gave a lower in vitro NH_3_ production than alfalfa hay as the low tannins’ counterpart. Quebracho extract addition into a diet of sheep wethers resulted in a lower ruminal NH_3_ and blood urea N concentrations [94]. Adding tannin extract from *Vaccinium vitis-idaea* into a diet of dairy cows decreased NH_3_ production [95]. Grainger et al. [89] concluded that tannin extracts from *Acacia mearnsii* barks in a diet of dairy cows reduced urinary N loss. A similar decrease in urinary N excretion was reported in wethers supplemented by a tannin extract from *Acacia mearnsii* [98]. Nevertheless, Puchala et al. [99] reported that there was no difference for NH_3_ productions between goats fed fresh *Sericea lespedeza,* rich in tannins, and those fed either alfalfa or sorghum–Sudan grass. A study comparing the growth stages of purple prairie clover, between vegetative and flowering stages with different CT contents, showed that they were not different in in vitro rumen NH_3_ production [84].

Tannins can also decrease rumen CH_4_ output by reducing the inter-species transfer of H_2_ into methanogenic bacteria, and hence depressing their growth [6,76,85]. Huang et al. [85] informed that CT extract supplementation from *Leucaena leucephala* reduced in vitro rumen GP and CH_4_ releases. Moreover, tannin extract addition from *Acacia mearnsii* into a diet of dairy cows reduced CH_4_ production [89]. It was similarly reported that goats fed either fresh *Sericea lespedeza,* rich in tannins, or its hay produced less CH_4_ in comparison with those fed either alfalfa or sorghum–Sudan grass [99]. However, Guglielmelli et al. [86] reported that Sainfoin hay released higher in vitro CH_4_ from the rumen than alfalfa hay.

Sainfoin hay supplementation produced higher rumen in vitro VFA and acetate, but no difference was reported in the acetate:propionate (A:P) ratio compared with alfalfa hay [86]. Wood et al. [83] found that *Chrysanthemun coronarium* supplementation likely acted to increase acetate but reduce propionate. Nonetheless, Cieslak et al. [95] reported that adding tannin extracts from *Vaccinium vitis-idaea* in a diet of dairy cow had no effect on VFA, but reduced the A:P ratio in the rumen fluid.

It was reported that CT extract supplementation from *Leucaena leucephala* had no impact on IVDMD, except for it being lower for the high dose [85]. An in vitro experiment comparing the growth stage of purple prairie clover between vegetative and flowering stages (58.6 and 94.0 g CT/kg DM, respectively) indicated that the vegetative stage had a higher IVDMD than flowering stage [84]. An in sacco investigation by Azuhnwi et al. [87] found that adding condensed tannins from sainfoin (*Onobrychis viciifolia* Scob) into a diet of dairy cow reduced DMI and CP degradability. Meanwhile, Guglielmelli et al. [86] reported that Sainfoin hay resulted in greater IVOMD than that by alfalfa hay. 

Kozloski et al. [98] indicated that adding tannin extract from *Acacia mearns* to a diet of sheep wethers resulted in a lower DMI and the digestibility of DM, OM, neutral detergent fiber (NDF), and N. Grainger et al. [89] also showed a reduction in DMI and milk yield in dairy cows supplemented with tannins extracted from *Acacia mearnsii*. Different things were presented by Costa et al. [97], in which the addition of *Acacia mearnsii* up to 40 g CT/kg (*Acacia mearnsii* contains 700 g CT/kg) in the lamb feed could increase nutrient intake and digestibility, as well as increase growth and feed efficiency. However, Briceño-Poot et al. [88] reported that the addition of *Acacia pennatula* or *Enterolobium cyclocarpum* into a diet of sheep resulted in a higher DMI, especially for those supplemented with *Acacia pennatula*. Similarly, it was reported that goats fed either fresh *Sericea lespedeza* or its hay had higher DMI but lower DM and N digestibility in comparison with those fed either alfalfa or sorghum–Sudan grass [99]. Owens et al. [93] informed that adding quebracho tannin extract from *Aspidosperma quebracho* into a diet of lambs resulted in no impact on DMI, digested DM, digested energy, or digested NDF, but increased N digestibility. Galicia-Aguilar et al. [100] reported that sheep supplemented by *Havardia albicans* had a similar DMI but lower DM digestibility. Cieslak et al. [95] observed that adding tannin extract from *Vaccinium vitis-idaea* into a diet of dairy cows had no impact on milk production and its fat, CP, lactose, and energy contents, as well as DM, OM, and NDF digestibility. In addition, adding quebracho tannins extract into a diet of sheep increased cis9, trans11 CLA (conjugated linoleic acid, rumenic acid) and polyunsaturated fatty acids (PUFA), but reduced saturated fatty acids (SFA) in the longissimus muscle [92] and increased vaccenic acid (trans11 C18:1) with no effect on stearic acid (C18:0) compositions in the rumen fluid [91]. 

Tannin addition into ruminant diets increased the rumenic acid and PUFA and decreased SFA in ruminant products, such as milk and meat, via modified bio-hydrogenation by altering the rumen microbial population [83,91,92]. Tannin supplementation, however, is thought to be associated with reduced feed intake, resulting in possible reduced nutrient intakes, digestibility, animal performance. These responses may be due to the possible toxicity of tannin-containing diets to animals [76,101].

#### 3.2.2. Effect of Tannins as Anthelmintics on Ruminants

Azaizeh et al. [80] reported that the *Pistachia lentiscus* and *Phillyrea latifolia* extracts inhibited the exsheathment of gastro-intestinal nematode larvae in vitro, while sheep supplemented with *Havardia albicans* had less *Haemonchus contortus* in their faeces [100]. Julaeha et al., [3] found that adding *Jatropha multifida* leaves into a diet of lambs reduced *Trichostrongylus spp.* fecal eggs counts. Tannins have the potential to increase animal health via their antioxidant properties and to prevent bloat as well as to break protein-rich cells of nematodes [102]. 

The other ruminant studies in vivo showed that tannins had the anthelmintic potentials. Saratsi et al. [103] stated that *Ceratonia siliqua,* rich in CT, had an anthelmintic effect. The cashew apple fiber added into a sheep’s diet as a source of tannins showed 40.8% effectiveness as an anthelmintic compared to a monepantel anthelmintic [104]. The other herbal plants tested in vivo such as green tea, oak leaves, and mixed herbs showed their effects on increasing host resistance to parasites, reducing the number of parasites, and increasing livestock productivities [105,106,107].

**Table 6 animals-13-00767-t006:** Effect of tannins as possible anthelmintics on different ruminant animals.

No	Tannins	Test Systems	Outputs	References
1.	*Havardia albicans* (71.5 g/kg DM CT) and basal diet (40:60, DM basis)	In vivo sheep fed grain-based concentrate and *Pennisetum purpureum* grass (90:10, DM basis)	No difference for DMI but lower DM digestibility; decreased *Haemonchus contortus* and females’ fecundity	[100]
2.	*Jatropha multifida* leaf powder (34.5% phenols, 13.2% tannins) at 0, 0.5, 0.75, and 1%	In vivo Lambs fed by Elephant grass: concentrate (80:20)	Reduced *Trichostrongylus* spp. fecal eggs counts and increased ADG at 0.5% inclusion optimally.	[3]
3.	*Pistachia* lentiscus and *Phillyrea latifolia* extracts (100% ethanol, 70% ethanol, or water extractions) at 1200 µg/mL of phosphate-buffered saline solution incubated with gastro-intestinal nematodes	Larval ex-sheathment inhibition assays (LEIA) with *Teladorsagia circumcincta, Teladorsagia colubriformis*, and *Chabertia ovina* (originally cultured from a donor goat)	Inhibited the ex-sheathment of gastro-intestinal nematode larvae for all extraction methods	[80]
4.	Carob (*Ceratonia siliqua*) pods, Sainfoin (*Onobrychis viciifolia*) pellets	In vivo lambs fed diets containing with or without tannin sources. Experiment 1: Carob meal (0, 3, 6, 9, and 12% of total diet). Experiment 2: 12% Carob meal in the diet. Experiment 3: (1) 12% Carob meal; (2) 35% sainfoin; (3) a combination of 12% carob and 35% sainfoin; (4) control (lucerne)	Carob-containing CT had an anthelmintic effect, but there was no clear indication of a synergistic effect with sainfoin	[103]
5.	Hydrolysable tannin (HT) extract from chestnut tree (*Castanea sativa*) at 0, 2, 4, 8, 25, and 50 mg/mL during 0.5, 1, 2, and 24 h.	In vitro with naturally infected sheep	The 25 mg/mL extract of hydrolysable tannins from chestnut was lethal for adults of *Haemonchus contortus.* HT can be an alternative nematode control in ruminants	[108]
6.	Cashew apple fiber (*Anacardium occidentale*): (1) control (no treatment), (2) anthelmintic monepantel 2.5 mg/kg PV, and (3) 0.3% BW cashew apple fiber	In vivo sheep fed corn silage	The cashew apple fiber showed 40.8% efficacy to destroy *Haemonchus contortus*, while anthelmintic monepantel was 99.6%	[104]
7.	*Elephantorrhiza elephantine* of ethanol, methanol, and water extracts.	In vitro naturally infected goat by *Paramphistomum cervi*	*Elephantorrhiza elephantine* had efficacy in controlling goat nematodes	[109]
8.	*Pistacia lentiscus, Phillyrea latifolia, Inula viscosa* clipped on winter, spring, summer, and fall at different concentration of 600, 900, 1200, and 2400 ppm.	LEIA with *Teladorsagia circumcincta* and *Trichostrongylus colubriformis*	Seasonal variations should be taken into account when plants are integrated into anthelmintic strategies.	[110]
9.	Mix herbs (8.55% each of *Althaea officinalis, Petasites hybridus, Inula helenium, Malva sylvestris, Chamomilla recutita, Plantago lanceolata, Rosmarinus officinalis, Solidago virgaurea*, *Fumaria officinalis*, *Hyssopus officinalis* and *Melissa officinalis*, 5% *Foeniculum vulgare* and 1% *Artemisia absinthium*)	In vitro and in vivo lambs fed meadow hay (600 g DM/day) and a concentrate (350 g DM/day; 70% barley, 22% soybean meal, 4.8% wheat bran, 0.5% bicarbonate, and 2.7% mineral–vitamin premix)	The combination of these different botanical family herbs contributed to slowing the dynamics of *Haemonchus contortus* infection and improved the production indicator of the lambs	[105]
10.	Green tea polyphenols (GTP) at 2, 4, and 6 g/kg feed	In vivo lambs fed *Aneurolepidium chinense* and grain-based concentrate (30:70)	Dietary GTP improved host resistance to *Haemoncus contortus* infection by reducing worm burdens and weight loss	[106]
11.	Oak leaves. Species 1: *Quercus semecarpifolia* (QS), species 2: *Quercus leucotricophora* (QL)	In vivo goats fed Concentrate: roughage sources (30:70) Roughage sources: *Pennisetum clandestinum*, QS, and QL	Reduced the gastrointestinal nematodes. It had a beneficial impact on augmenting nutrient utilization, growth performance and feed efficiency. Goats fed QS-based diet showed better performance compared with those fed QL-based diet	[107]
12.	Ethanol extract of *Inula viscosa, Salix alba,* and *Quercus calliprinos* at 600, 1200, 2400 ppm	In vitro developmental assay of *Haemonchus bacteriophora* population reared in the late-instar larvae of *Galleria mellonella*	Plant extracts were highly toxic to the survival of the eggs and young juveniles at all concentrations. The extracts inhibited their development, associated with low reproduction parameters.	[111]

Acevedo-Ramírez et al. [108] conducted a sheep in vitro study using tannins derived from a chestnut tree. The results indicated that tannins can cause the death of adult *Haemonchus contortus*, so that tannins can be used as an alternative to conventional nematode control agents in ruminants. This is similar to the results reported by Mazhangara et al. [109], who tested tannins in *Elephantorrhiza elephantine*. Studying tannins as anthelmintics was also carried out using the larval ex-sheathment inhibition assay (LEIA), where *Pistacia lentiscus, Phillyrea latifolia*, and *Inula viscosa,* harvested in different seasons, showed different anthelmintic effectiveness. Azaizeh et al. [110] and Santhi et al. [111] tested an ethanol extract of *Inula viscosa, Salix alba*, and *Quercus calliprinos* using an in vitro developmental assay, which showed that these tannin-rich plant extracts were considerably toxic to the eggs and larvae of *Heterorhabditis bacteriophora.*

Tannins can act as an antiparasitic agents in ruminants. The efficacy of tannins in reducing gastrointestinal nematodes is by increasing the host response to parasites. The capability of tannins to bind to proteins is able to protect them from rumen degradation, and improve protein flow and amino acid absorption in the small intestine [3]. Increased protein supply in the small intestine is seen to enhance host homeostasis and immune response to helminths [12].

### 3.3. Saponin Sources, Types, and Uses

Saponins are distributed in most parts of the plant, such as the leaves, seeds, roots, tubers, and tree bark. Some plant sources that contain tannins are *Camelia sinensis* var. Assamica, *Dioscorea pseudojaponica* Yamamoto, and *Quillaja saponica*. All of these plants have saponins in various forms, as described in more details in Table 2. Saponins are a diverse unit of low-molecular-weight, plant-bioactive compounds. Saponins have the capability to form stable soap-like foams in watery solution.

**Table 7 animals-13-00767-t007:** Chemical characteristics of saponins in different botanical parts of some saponin-rich plants.

Plants	Scientific Names	Main Parts	Major Bio-Active Compounds	References
Chinese chive	*Allium tuberosum*	Seeds	26-*O*-*β*-D-glucopyranosyl-(25*S*,20*R*)-20-*O*-methyl-5*α*-furost-22(23)-en-2*α*,3*β*,20,26-tetraol 3-*O*-*α*-L-rhamnopyranosyl-(1→2)-[*α*-L-rhamnopyranosyl-(1→4)]-*β*-D-glucopyranoside, 26-*O*-*β*-D-glucopyranosyl-(25*S*,20*R*)-5α-furost- 22(23)-en-2*α*,3*β*,20,26-tetraol 3-*O*-*α*-L-rhamnopyranosyl-(1→2)-[*α*-L- rhamnopyranosyl-(1→4)]-*β*-D-glucopyranoside; 26-*O*-*β*-D-glucopyranosyl-(25*S*,20*S*)-5*α*-furost- 22(23)-en-2*α*,3β,20,26-tetraol 3-*O*-*α*-L- rhamnopyranosyl-(1→2)-[*α*-L- rhamnopyranosyl-(1→4)]-*β*-D-glucopyranoside, 26-*O*-*β*-D-glucopyranosyl-(25*S*,20*S*)-5*α*-furost- 22(23)-en-3*β*,20,26-triol 3-*O*-*α*-L- rhamnopyranosyl-(1→2)-[*α*-L- rhamnopyranosyl-(1→4)]-*β*-D-glucopyranoside	[112]
Tea	*Camelia sinensis* var. Assamica	Roots	Triterpenoid saponin structures: methyl esters of 3-O-α-L-arabinopyranosyl (1→3)-β-D-glucuronopyranosyl-21, 22-di-O-angeloyl-R1-barrigenol-23-oic acid, 3-O-α-L-arabinopyranosyl (1→3)-β-D-glucuronopyranosyl-21-O-angeloyl-22-O-2-methylbutanoyl-R_1_-barrigenol-23-oic acid, 3-O-α-L-arabinopyranosyl, (1→3)-β-D- glucuronopyranosyl-16α-O-acetyl-21-O-angeloyl-22-O-2-methylbutanoyl-R_1_-barrigenol-23-oic acid	[113]
Yam	*Dioscorea pseudojaponica* Yamamoto	Tubers	(Steroidal sapoinins) methyl protodioscin and methyl protogracillin (furostanol glycosides), dioscin and gracillin (spirostanol glycosides). Their structures: 26-*O*-β-D-glucopyranosyl-22α- methoxyl-(25*R*)-furost-5-en-3β,26-diol, 3-*O*-α-L-rhamnopyranosyl-(1→2)-*O*-[[α-L rhamnopyranosyl-(1→4)]-*O*-[α-L-rhamnopyranosyl-(1→4)]]-β-D-glucopyranoside; (25*R*)-spirost-5-en-3β-ol 3-*O*-α-L-rhamnopyranosyl-(1→2)-*O*-[[α-L-rhamnopyranosyl-(L→4)]-*O*-[α-L-rhamnopyranosyl-(1→4)]]-β-d-glucopyranoside	[114]
Quillaja	*Quillaja saponaria*	Barks	Triterpenoid saponin sturctures: 3-O-[β-D- galactopyranosyl-(1→2)-[3-O glucopyranosiduronic acid], 3-O-[α-L-rhamnopyranosyl-(1→3)-[β-D- galactopyranosyl-(1→2)]-β-D- glucopyranosiduronic acid], *3-O-*[[*β-* D-xylopyranosyl-(1→3)-[β-D-galactopyranosyl-(1→2)]-[3-O-glucopyranosiduronic acid].	[15]

Chemically, saponins comprise a sugar moiety, commonly containing glucose, galactose, glucuronic acid, xylose, rhamnose, or methyl pentose, which is glycosidically related to a hydrophobic aglycone (sapogenin) in the form of either triterpenoids or steroids [5,115]. Triterpenoids are widely distributed in nature in comparison with steroids [116]. The usual form of triterpenoid aglycone is a derivative of oleanane, while the main forms of steroid aglycones are mostly found in the spirostanol and furostanol derivatives [115,116]. The aglycone may consist of one or more unsaturated C-C bonds [5]. The chain of oligosaccharides is commonly attached at the C3 location (monodesmosidic), but there are numerous saponins found to have an extra sugar moiety at the C26 or C28 positions (bidesmosidic) [116]. Wina et al. [115] also reported that there were two general types of triterpenoid saponins: neutral and acidic. Neutral saponins have their sugar components attached to sapogenin, while acidic saponins have their sugars moiety containing uronic acid, or with one or more carboxylic units attached to the sapogenin [115].

#### 3.3.1. Effects of Saponins as Dietary Additives on Ruminants

Several studies have shown that tea saponins have a suppressing impact on the release of CH_4_ and NH_3_ in vitro [117] and in vivo by using growing lambs [118]. The CH_4_ reduction was supported by the reduction in protozoa and particularly the protozoa-related methanogens [115,119]. Saponins can act as defaunation agents via a sterol–saponin interaction in the protozoal cell membrane, hence affecting the methanogenic protozoa [115]. Since protozoa can be a predator for bacteria, at an appropriate level, defaunation may improve the population of bacteria and may increase N utilization, leading to improved animal growth and meat or milk productions [115]. Less protozoa in the rumen is also likely to result in less acetate production, since most fermentation end products of protozoa comprise acetate [6,115].

**Table 8 animals-13-00767-t008:** Effect of different saponins as feed additives on ruminants.

No	Saponins	Basal Diets	Test Systems	Outputs	References
1.	Saponins extract from *Achyranthus aspara, Tribulus terrestris* and *Albizia lebbeck* at 3, 6, or 9% in the substrate (DM basis)	Wheat straw and concentrate (50:50)	In vitro buffalo	Decreased CH_4_, from (ml/mg DM) 37.5 (control) to 19.2–24.5; decreased protozoa and NH_3_; no effect on IVDMD and tVFA but A:P ratio tended to decrease	[120]
2.	Saponins extract from *Gynostemma pentaphyllum* (98% gynosaponin, Kangwei Bioengineering Ltd., China) at 50, 100, or 200 mg/L medium	A mixed co- culture medium of anaerobic fungi and methanogens from goat rumen contents	In vitro goat	Reduced tGP, CH_4_, tVFA, fungi, and methanogens but increased pH at increased levels of saponin addition	[121]
3.	Waru leaf (*Hibiscus tiliaceus*) at 5, 10, 15, or 20% saponins in substrate to equally substitute Napier grass	Napier grass (*Pennisetum purpureum*)	In vitro cattle	Decreased tGP, in line with increased saponin levels; tended to increase tVFA at 5 and 10% saponin levels; no difference for A:P, but it tended to decrease linearly at increased saponin levels; no effect on pH and NH_3_; reduced protozoa with the lowest at 5%.	[122]
4.	Saponins extract from Agave aloe (AE, *Agave Americana*) at 120, 240, or 360 mg saponins/kg DMI and *Quillaja saponaria* (QS) at 120 mg saponins/kg DMI	Oaten hay (ad libitum), barley-based concentrate (400 g/sheep/d)	In vivo lambs	No effect on DMI, nutrient intake, OM, CP, and NDF digestibility, or N balance, but reduced protozoa number in RF, blood cholesterol and glucose; tended to increase ADG (g/d) (59.6 for control vs. 77.8, 77.2, 79.0, and 76.6 for AE at 120, 240, 360 and QS at 120 mg saponins/kg DMI	[123]
5.	Tea saponins extract from green tea leaves (*Ilex kudingcha* C.J. Tseng, >70% triterpenoid saponins) at 0.4, 0.6, and 0.8 g total saponins/kg DM	Maize stover (forage) and concentrate (50:50)	In vivo goats	No effect on DM, N, or ADF intakes; no effect on DM, N, or ADF digestibility, either in rumen or small intestines; no effect on amino acid digestibility in small intestine; no effect on rumen pH, VFA, A:P, or NH_3_	[124]
6.	Saponins extract from *Quillaja saponaria* (Sigma-Aldrich Inc., St. Louis, MO, USA) at 20 g saponins/kg diet	Beet-pulp-based diet containing alkaloids: gramine at 2 g/kg or methoxy-N,N dimethyltryptamine at 0.03 g/kg diet	In vivo lambs	No effect on tDMI, total digested DM, energy, N, or NDF	[93]
7.	*Yucca schidigera* steroidal-rich saponins extract (YS) (from stems, 8.5% saponins, Desert King International, San Diego, CA, USA), *Quillaja saponaria* triterpenic-rich saponin extract (QS, from barks tree, 3.6% saponins, Desert King International, San Diego, CA, USA) or *Camellia sinensis* triterpenic-rich saponin extract (TS, from whole plant, 21.6% saponins, Ningbo Good Green Sci. and Tech., Ningbo, China) at 1.5, 0.64, or 0.25% saponins in DM of diets, respectively	Corn- and corn-silage-based diet	In vivo steers	YS and QS showed no differences compared tocontrol for DMI and ADG, but N intake of YS was lower than control and QS; TS had higher DMI and N intake but had a similar ADG to the control; no effect on DM, NH_3,_ and N of daily manure excretion; TS had lower NH_3_ than control; No effect on CH_4_ in general, but increased TS inclusions, from 0.25% to 0.5%, resulted in CH_4_ decreasing by 31%, and reducing DMI and ADG	[125]
8.	Tea saponins extract (> 60% triterpenoid saponins, Zhejiang Orient Tea Development Co., Ltd., China) at 3 g/lamb/d	Chinese wild rye grass and concentrate (60:40)	In vivo lambs	No effect on feed intake and daily gain; reduced CH_4_ (L/kg DMI); increased tVFA but no effect on A:P; decreased ruminal pH and reduced NH_3_; no effect on methanogens, fungi, *R. flavefaciens,* orand *F. succinogenes,* but decreased protozoa populations. Reduced SFA, *cis*9, *trans*11 CLA/vaccenic acid ratio; increased MUFA, but no effect on PUFA (*longissimus dorsi* muscle)	[118,126]
9.	Saponins extract from barks of *Quillaja saponaria* (Sigma Batch: 024K2505, Santiago, Chile, USA) at 6, 12, and 18 mg sapogenin/kg DMI	Ad libitum Oat hay and barley-based concentrate (400 g/lamb/d)	In vivo lambs	No effect on the intakes of DM, OM, CP, or NDF, or the digestibility of DM, OM, or CP, but decreased NDF digestibility; no effect on N balance, N supply, pH, or NH_3_ but decreased protozoa numbers and glucose; no effect on ADG, cooking loss, or meat pH (24 h post mortem), but decreased carcass weight	[127]
				Reduced the concentration of *cis*9 C14:1 (*longissimus dorsi* muscle) and its desaturation index; 12 mg had higher C20:4n6 than control and 6 mg; 12 mg had lower α-linolenic:linoleic ratio than control; no effect on muscle cholesterol levels	[10]
10.	*Acacia concinna* pods (5.0 g saponins/kg DM; *Syzygium aromaticum* buds EO 2.5 g saponins/kg DM. Both plants added to the concentrate as premix	Concentrate mixture containing sunflower oil (66.7 g/kg DM): berseem hay (60:40)	In vivo goats	*Acacia concinna* had no influence on FA composition in muscle and adipose tissues. *Syzygium aromaticum* has the potential to enhance the health-promoting VA and *cis-9*, *trans-*11 CLA concentrations in the meat of goats	[128]
11.	*Quillaja saponaria* (0.6 and 1.2 g saponins/L); propionate (4 and 8 mM); nitrate (5 and 10 mM). Treatment consisted of single doses and combination of all	Rumen donor cows fed corn silage (45%); alfalfa hay (10%); Cargill dairy protein product (20%), and concentrate mixture (25%)	In vitro cow	Saponins and nitrate substantially decreased CH_4_ and methanogens in an additive manner. Saponin and nitrate, in combination, improved feed digestion and rumen fermentation	[45]
12.	Tea saponins/TSP (0–0.50 g/L and 0.52% TSP in DM diets	54% corn silage, 6% hay, and 40% pelleted concentrate	In vitro and in vivo dairy cows	Tea saponins reduced lactation performance and DMI. The 0.52% DM plant extract failed to reduce daily CH_4_ production. Tea saponin is not efficient to reduce methane emissions from dairy cows.	[129]

Goel and Makkar [130] reported, in vitro, that adding saponin extracts from either *Achyranthus aspara, Tribulus terrestris*, or *Albizia lebbeck* at 3, 6, or 9% dietary DM decreased CH_4_ by 34–48%. Wang et al. [121] reported, in vitro, that adding saponin extracts from *Gynostemma pentaphyllum* (98% gynosaponin) at 50, 100, or 200 mg/L medium of a mixed co-culture of anaerobic fungus and methanogens from goat rumen contents reduced GP and CH_4_ production. It was also reported that waru leaf (*Hibiscus tiliaceus*) additions at 5, 15, or 20% saponin levels into a Napier grass (*Pennisetum purpureum*)-based diet decreased GP linearly [122]. Similarly, an in vivo lamb investigation by Mao et al. [118] found that adding tea saponin extract (>60% triterpenoid saponins) at 3 g/lamb/day reduced CH_4_ production by about 27%. However, Li and Powers [125] indicated in vivo that adding either *Yucca schidigera, Quillaja saponaria*, or *Camellia sinensis* extracts at 1.5, 0.64, or 0.25% saponin content, respectively (DM basis), into a corn- and corn-silage-based diet had no impact on CH_4_ output per unit of DMI in steers.

Goel and Makkar [130] reported that adding saponin extract reduced NH_3_ production, but Istiqomah et al. [122] found in vitro that waru leaf supplementation had no effect on NH_3_ production. Although Mao et al. [118] reported that adding tea saponin extract into a diet tended to reduce NH_3_ production (143.0 vs. control, 167.5 mg/L), Zhou et al. [124] observed in vivo that green tea saponin extract additions at 0.4, 0.6, or 0.8 g saponins/kg dietary DM had no effect on the NH_3_ production of goats. Similarly, Nasri et al. [127] found in vivo that adding saponins extract from *Quillaja saponaria* at 6, 12, or 18 mg sapogenin/kg dietary DM had no effect on the NH_3_ production of the lambs.

It was reported in vitro that waru leaf inclusions into a Napier grass-based diet were likely to increase VFA, but Wang et al. [121] reported in vitro that saponin extract supplementation from *Gynostemma pentaphyllum* reduced VFA without affecting VFA proportions. Mao et al. [118] observed in vivo that adding tea saponin extract into a diet of lambs increased VFA with no effect on the A:P ratio, while Zhou et al. [124] observed that green tea saponin extract inclusions had no effect on either the tVFA or A:P ratio in the rumen liquid of goats.

Wang et al. [121] found in vitro that adding saponins extract from *Gynostemma pentaphyllum* increased ruminal pH, but Istiqomah et al. [122] found in vitro that waru leaf addition in Napier grass resulted in no impact on ruminal pH. An in vivo lamb study by Mao et al. [118] reported that adding tea saponin extract into a diet decreased ruminal pH, but Zhou et al. [124] observed in vivo that green tea saponin extract supplementation had no impact on ruminal pH in goats. Similarly, Nasri et al. [127] found in vivo that saponin extract supplementation from *Quillaja saponaria* into oat hay- and barley-based diets had no impact on ruminal pH in lambs.

It was observed in vitro that saponin extract inclusions from either *Achyranthus aspara, Tribulus terrestris* or *Albizia lebbeck* had no effect on IVDMD [130]. Meanwhile, an in vivo study by Nasri and Ben Salem [123] found that adding saponin extract from *Agave Americana* at 120, 240, or 360 mg saponins/kg and *Quillaja saponaria* at 120 mg saponins/kg dietary DM had no effect on DMI and nutrient intakes, and no effect on the OM, CP, and NDF digestibility of lambs. Similarly, Owens et al. [93] reported that adding saponin extracts from *Quillaja saponaria* at 20 g saponins/kg (Beet pulp-based diet containing alkaloids, either gramine at 2 g/kg or methoxy-N, N-dimethyltryptamine at 0.03 g/kg diet) had no effect on DMI or the total digested DM, energy, N, and NDF by lambs. Mao et al. [118] studied in vivo that tea saponin extract inclusions had no impact on feed intakes and weight gain of lambs. Zhou et al. [124] studied in vivo that green tea saponin extract supplementation had no impact on the intakes and the digestibility of DM, N, and ADF of goats. Li and Powers [125] added either *Yucca schidigera* (YS), *Quillaja saponaria* (QS) or *Camellia sinensis* extracts (TS, tea saponins) into a corn- and corn-silage-based diet of steers, and found that QS and YS had no difference compared with the control diet in DMI and ADG, but the N intake of YS was lower than the control diet and QS, while TS had higher DMI and N intake but having a similar ADG to the control diet. In addition, it was reported in vivo that adding saponin extracts from *Quillaja saponaria* at 6, 12, and 18 mg sapogenin/kg DMI in an oat hay- and barley-based diet had no effect on the intakes of DM, OM, CP, and NDF, the digestibility of DM, OM, and CP, as well as ADG, cooking loss, and meat pH, but decreased NDF digestibility in lambs [127]. Brogna et al. [10] also found a reduction in the concentration of C14:1 cis-9 from the longissimus dorsi muscle and its desaturation index, increased C20:4n-6, and decreased α-linolenic:linoleic ratio at a saponin level of 12 mg, with no effect on muscle cholesterol concentrations of lambs. Meanwhile, Mao et al. [126] reported that adding tea saponin extracts (>60% triterpenoid saponins) into a diet of lambs reduced SFA and the rumenic:vaccenic acid ratio and increased MUFA, but it had no effect on PUFA in the longissimus dorsi muscle.

Another study was conducted in vitro by Mandal et al. [128] and Patra and Yu [45] on goats and cattle, respectively. *Acacia concinna* showed no impact on fatty acid conformation in muscle and adipose tissue, while *Quillaja Saponaria* showed its effect on decreasing CH_4_ production. On the other hand, the use of saponins from tea leaves in vitro and in vivo in dairy cows has shown its effect on decreasing lactation performance and dry matter consumption, but did not reduce CH_4_ production, so that saponins in tea are considered inefficient to reduce CH_4_ emissions [129].

#### 3.3.2. Effect of Saponin as Anthelmintics on Ruminants

Botura et al. [131] reported, in vivo, that supplementing either sisal waste extract (SWE) (*Agave sisalana*, containing hecogenin and tigogenin) at 1.7 g/goat/day or levamisole phosphate (LEP) (6.3 mg/kg) as a positive control into grass hay-fed goats reduced fecal egg counts by a maximum of 50.3% (SWE) and 93.6% (LEP). In this study, LEP reduced the recovered parasites from the digestive tract by 74%, but a small decrease of parasites was reported for SWE. There was no toxicity effect reported from both treatments, as measured by the histological analysis of the kidney and liver. Another experiment was carried out using the egg hatch assay (EHA) and larval migration inhibition (LMI) methods using *Phytolacca icosandra* [132] and *Agave sisalana* (aqueous extract) [133]. Both studies showed that *Phytolacca icosandra* in ethanol and dichloromethane extracts could destroy *Haemonchus contortus* eggs and larvae, while the saponins contained in *Agave sisalina* could also attack nematodes in the digestive tract of ruminant animals.

**Table 9 animals-13-00767-t009:** Effect of different saponins as possible anthelmintics on ruminants.

No	Saponins	Test systems	Outputs	References
1.	Sisal waste extract (SWE) (*Agave sisalana,* containing saponins in the form of sapogenins hecogenin and tigogenin) at 1.7 g/goat/day; levamisole phosphate (LEP) (6.3 mg/kg) as a (+) control	In vivo goats fed by grass hay	Reduced fecal egg count by max. 50.3% (SWE) and 93.6% (LEP); LEP reduced the recovered parasites from the digestive tract by 74% but a low decrease of those parasites for SWE. No toxicity effect from both treatments assessed by histological analysis of the liver and kidney	[131]
2.	*Phytolacca icosandra* (ethanol, *n-*hexane, and dichloromethane extract) at 0.5, 1.0, 2.0, 3.0, and 4.0 mg/mL. (+) control: Thiabendazole (EHA). (−) control: Tween*^®^* 80 (LMIA), untreated egg in water (EHA)	In vitro LMI and EHA assays by a donor sheep with a monospecific infection of *Haemonchus contortus*	Saponins were only found in the ethanolic extract of *Phytolacca isocandra.* Ethanolic and dichloromethane extracts of the plants showed in vitro anthelmintic activity against the *H. contortus* eggs and the L_3_ larvae. However, the hexanic extract of the plant leaves failed to show any in vitro anthelmintic activity.	[132]
3.	*Agave sisalana* in the form of an aqueous extract (AE), ethyl acetate extract (EE), flavonoid fractions (FF), and saponin fraction (SF). EHA treatment: AE: 0.625, 1.25, 2.5, 5, and 10 mg/mL; EE and FF: 0.02, 0.04, 0.08, 0.16, and 0.32 mg/mL; SF: 0.32 mg/mL; (−) control: distilled water; (+) control: albendazole 0.025 mg/mL. LMI treatment: AE and EE: 100 mg/mL; FF and SF: 2.5 mg/mL; (−) control: PBS; (+) control: levamisole (0.5 mg/mL)	In vitro EHA and LMI with naturally infected goats (fecal culture 81% *Haemonchus* spp., 14% *Oesophagostomum*, and 5% *Trichostrogylus* spp.)	The saponin fractions showed no ovicidal activity while flavonoid fractions did not show activity against larvae. *Agave sisalana* was active against the gastrointestinal nematodes of goats, related to the presence of homo-isoflavanoid saponin compounds.	[133]
4.	*Elephantorrhiza elephantina* roots (83.28 ± 1.72% saponins) in ethanol, methanol, and water extract at 1.87, 3.75, 7.5, and 15 mg/mL	In vitro adult motility inhibition assay with naturally infected goats	Ethanol, methanol, and water extract of *Elephantorrhiza elephantina* roots showed a potential anthelmintic activity against adult *Paramphistomum cervi* worm motility, in botha a time- and dose-dependent mannerand.	[109]

The hatching process of nematode eggs begins with a stimulus from the environment, which causes the larvae to release several enzymes, such as proteases, lipases, and chitinases, that function to degrade the egg membrane [133]. The flavonoid compounds contained in *Agave sisalana* can inhibit the activity of these enzymes, so that changes in enzyme activity interfere with the egg-hatching process, resulting in the destruction of infectious worms [133].

## 4. Conclusions and Perspectives

Essential oils, polyphenols, and saponins are plant secondary metabolites found in various type of plants that can be extracted from different botanical parts of many plants. These materials can function as dietary additives and anthelmintics to increase the production and health performance of ruminants. Each bioactive constituent has a specific function and efficacy to achieve pre-defined objectives. However, the literature shows that these compounds may be variable in their effectiveness depending upon the plant sources, extraction methods, amounts, and diets in various studies in different situations. Several bioactive-compound-based dietary supplements can reduce methane or nematode parasites. However, these great reductions are sometimes followed by significant declines in feed intake and performance of the animals. Therefore, it is essential to select the most appropriate plants that contain compounds selected for their appropriate dosages and applications, to either optimize rumen function or reduce methane and anthelmintics in a range of ruminant animals. Moreover, it is vital to test the potential safety, affordability and efficiency of their dietary inclusion for ruminant animals, and, consequently, the impacts on consumers, and the environment.

## Figures and Tables

**Figure 1 animals-13-00767-f001:**
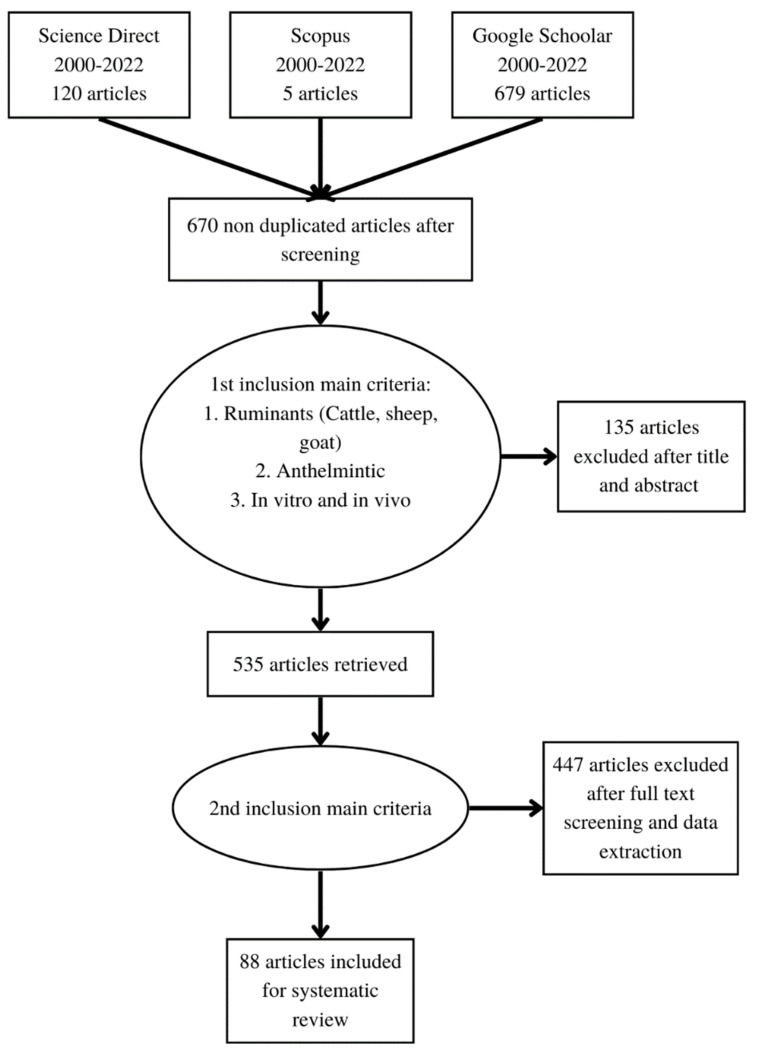
Flow chart of PRISMA protocol that was followed in the current systematic review.

## Data Availability

No new data were created or analyzed in this study. Data sharing is not applicable to this article.

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
