# Peer review of "Roles of Essential Oils, Polyphenols, and Saponins of Medicinal Plants as Natural Additives and Anthelmintics in Ruminant Diets: A Systematic Review"

_animals, 2023, doi:10.3390/ani13040767_

Round 1

Reviewer 1 Report

The author should consider the below comments to improve the paper.

1.                    The authors should reformulate the abstract in order to emphasize the novelty of the paper and the relevance of your paper for the readership of the current journal. In addition the introduction section should be enriched with more data on what has been already published on this matter and what gaps do the authors fill with the current manuscript?

2.                    Section 3.1. should removed because presents data only on the composition of EO. Such information can be already found in the literature in papers that summarize the literature data on EO composition. The same comment for the introductive parts of section 3.2 and 3.3.

3.                    All the tables in the manuscript are too long and contain too much information. They should be reduced and formulated in more concise way. Maybe shorter tables and more discussions on the data presented initially in tables would be better.

4.                   Please insert a list of abbreviations which would be useful considering the notations used in the paper.

5.                   Conclusion section is too short and it needs to be consolidated with numerical findings as well.

Author Response

Please see the attached file below

Reviewer 2 Report

Formatting revision is strongly recommended (e.g. tables, line spacing, Latin words etc.).

Conclusion is need to completed as it is mentioned in the text.

Most of the errors, comments and recommendations are listed in the attachment.

Author Response

Please see the attached file below

Round 2

Reviewer 1 Report

Thank you! All the mentioned objectives have been corrected!